# Early Prognostic Stratification of *Clostridioides difficile* Infection in the Emergency Department: The Role of Age and Comorbidities

**DOI:** 10.3390/jpm12101573

**Published:** 2022-09-24

**Authors:** Marcello Covino, Antonella Gallo, Erika Pero, Benedetta Simeoni, Noemi Macerola, Celeste Ambra Murace, Francesca Ibba, Francesco Landi, Francesco Franceschi, Massimo Montalto

**Affiliations:** 1Department of Emergency Medicine, Fondazione Policlinico Universitario “A. Gemelli”, IRCCS, Largo A. Gemelli, 8, 00168 Rome, Italy; 2Department of Geriatrics and Orthopedics, Fondazione Policlinico Universitario “A. Gemelli”, IRCCS, Largo A. Gemelli, 8, 00168 Rome, Italy; 3Department of Geriatrics and Orthopedics, Università Cattolica del Sacro Cuore, Largo A. Gemelli, 8, 00168 Rome, Italy; 4Department of Emergency Medicine, Università Cattolica del Sacro Cuore, Largo A. Gemelli, 8, 00168 Rome, Italy

**Keywords:** *Clostridioides difficile*, aging, frailty, diarrhea, comorbidities

## Abstract

*Clostridioides difficile* infection (CDI) represents a significant cause of morbidity and mortality, mainly in older and frail subjects. Early identification of outcome predictors, starting from emergency department (ED) admission, could help to improve their management. In a retrospective single-center study on patients accessing the ED for diarrhea and hospitalized with a diagnosis of CDI infection, the patients’ clinical history, presenting symptoms, vital signs, and laboratory exams at ED admission were recorded. Quick sequential organ failure assessments (qSOFA) were conducted and Charlson’s comorbidity indices (CCI) were calculated. The primary outcomes were represented by all-cause in-hospital death and the occurrence of major cumulative complications. Univariate and multivariate Cox regression analyses were performed to establish predictive risk factors for poor outcomes. Out of 450 patients, aged > 81 years, dyspnea at ED admission, creatinine > 2.5 mg/dL, white blood cell count > 13.31 × 10^9^/L, and albumin < 30 µmol/L were independently associated with in-hospital death and major complications (except for low albumin). Both in-hospital death and major complications were not associated with multimorbidity. In patients with CDI, the risk of in-hospital death and major complications could be effectively predicted upon ED admission. Patients in their 8th decade have an increased risk independent of comorbidities.

## 1. Introduction

*Clostridioides difficile (C. difficile)* disease is one of the most common hospital-acquired infections, and the first cause of nosocomial diarrhea [1,2]. It represents a significant cause of morbidity and mortality among hospitalized patients [2].

The infection shows a high variability of clinical pictures, ranging from the state of asymptomatic carrier, mild or moderate diarrhea, to severe cases characterized by fulminant colitis, complicated by dehydration, shock, toxic megacolon, ileus, and colon perforation [1]. Several factors determine a higher risk of developing *C. difficile* infection (CDI), including increasing age, length of hospitalization, ongoing and previous antibiotic therapy, chemotherapy, HIV infection, gastrointestinal surgery, or procedures of the gastrointestinal tract, including enteral nutrition [2]. The role of proton pump inhibitor (PPI) therapy as a potential risk factor remains uncertain [2].

Aging has been recognized worldwide as one of the most relevant risk factors for the development of CDI; about 70–80% of CDI infections occur in adults aged 65 years and older [1]. A higher burden of multiple comorbidities, frequent antibiotic use, increased health care exposure, and biological changes in the immune system may contribute to the increased vulnerability to this infection with advancing age [3,4,5]. Although conceivable that elderly subjects are at a greater risk not only of developing the symptomatic disease, but also associated complications, including progression to severe or fulminant disease and recurrences, very few studies have evaluated the role of other prognostic risk factors besides age in older people.

Up to now, different scores have been proposed to discriminate between severe and non-severe CDI [6]. These are based on a combination of clinical, laboratory, and radiologic/endoscopic parameters [7,8,9,10]; the latter are sometimes difficult to obtain, thus leading to the questionable use of scores in clinical practice. As regards disease outcome, several factors have shown a predictive capacity for poor prognosis [7]. Patient age, treatment with systemic antibiotics occurring on 1 or more days of *C. difficile* therapy, high leukocyte count and serum creatinine, and low albumin were associated with poor outcomes defined by the need for colectomy or death within 30 days after *C. difficile* onset [11,12,13]. However, these studies only included patients admitted to internal medicine or gastroenterology departments, and little is known about the prognostic stratification in the early phases of the disease [11,12,13].

In particular, limited data are available on CDI in the emergency department (ED) setting. Usually, time spent in an ED is not enough to perform an accurate prognostic evaluation, and clinical judgment often represents the best, and the most practical, instrument to guide immediate strategies. However, considering the relevant burden of CDI on healthcare resources, and the actual risk of a rapid deterioration of clinical status, waiting several days for a more detailed characterization and stratification of these patients does not represent the best way to face the disease. Alternatively, the early identification of outcome predictors in the ED could help to improve management strategies and risk stratification for these patients.

This study aimed at evaluating the performance, in different age subgroups, of clinical and laboratory parameters, in the early identification of CDI patients who are at risk of poor outcomes.

## 2. Materials and Methods

### 2.1. Study Population and Design

This is a retrospective single-centric study conducted in a tertiary urban university hospital with annual attendance at the ED of about 75,000 patients (more than 87% adults). We evaluated the electronic clinical records of consecutive patients admitted to the ED for 5 years from January 2016 to December 2020.

We enrolled all patients that accessed the ED for the occurrence of diarrhea, received a diagnosis of CDI infection, and were subsequently hospitalized. According to international guidelines [2], diagnosis of CDI was based on the presence of diarrhea (defined as ≥3 unformed stools passed per day) and a positive *C. difficile* toxin A/B enzyme immunoassay, obtained within 3 days of admission to the ED, to exclude cases of “healthcare facility onset CDI” [2]. Subjects aged < 18 years and pregnant women were excluded.

### 2.2. Patient Findings and Clinical History

All clinical and demographic data were extracted anonymously from the hospital’s computerized clinical records. Patient data were reviewed to assess clinical and demographic data, major comorbidities, and main clinical symptoms at admission.

In particular, for all patients included in the analysis, we recorded:Demographic data, including age and gender. According to age, the patients were divided into three different subgroups: 18–64 years; 65–79 years; 80 years and older;Vital signs, including systolic and diastolic blood pressure, heart rate, respiratory rate, oxygen saturation, body temperature;Laboratory parameters, including white blood cell count, serum creatinine, and serum albumin;Main symptoms at ED admission;Vital parameters and clinical presentation at admission were used to calculate, for each patient, the quick sequential organ failure assessment (qSOFA) score [14]. According to this score, patients were divided into two groups: suspected sepsis at presentation (qSOFA ≥ 2), and non-septic presentation (qSOFA < 2);Comorbidities, including hypertension, severe obesity (defined as BMI > 40), history of coronary artery disease (CAD), congestive heart failure, cerebrovascular disease, dementia, diabetes, chronic obstructive pulmonary disease (COPD), chronic kidney disease, malignancy. Overall comorbidity was assessed via the Charlson comorbidity index (CCI) for each patient [15].

### 2.3. Outcome Measures

The primary outcome measure was represented by:All-cause in-hospital death;The occurrence of major complications, which included all-cause in-hospital death, admission to intensive care unit (ICU)/need for mechanical ventilation, sepsis occurrence [16], and surgical intervention.

### 2.4. Statistical Analysis and Sample Size

Continuous variables are reported as the median (interquartile range), and were compared via univariate analysis using Mann–Whitney U tests, or Kruskal–Wallis tests in cases of three or more groups. Categorical variables were reported as absolute numbers (percentage), and were compared using chi-square tests (with Fisher’s test if appropriate).

Receiver operating characteristic (ROC) curve analysis was used to evaluate the overall performance of the qSOFA score in predicting in-hospital death and major complication occurrence. Follow-up and length of hospital stay were calculated from the time of ED admission to discharge or death. Survival curves were estimated via the Kaplan–Meier method.

The study variables were assessed for associations with all-cause in-hospital death via a univariate Cox regression analysis. The variables that reached statistical significance in the univariate analysis were entered into a multivariate Cox regression model to identify independent risk factors for survival.

We categorized all the continuous variables into dichotomous parameters (i.e., low/high) for better model fitting. For each variable, we obtained the optimal dividing cutoff by Youden’s index, performing a ROC curve analysis concerning association with death. Multivariate models excluded the single items composing derived variables to avoid model overfitting and parameter overestimation.

The calculated risk of in-hospital death was expressed as a hazard ratio (HR) (95% confidence interval). A two-sided *p*-value of ≤0.05 was considered significant in all the analyses. Data were analyzed by SPSS v25^®^ (IBM, Armonk, NY, USA) and MedCalc v18^®^ (MedCalc Software Ltd., Ostend, Belgium).

### 2.5. Institutional Review Board Statement

The study was conducted following the 1975 Declaration of Helsinki, and its later amendments, and was approved by the local Institutional Review Board (IRB #005181419). Each patient provided informed consent to be included in the analysis.

## 3. Results

During the study period, a total of 634 patients were admitted to our ED for CDI and subsequently hospitalized. Among them, 122 patients not meeting the inclusion criteria and 62 patients having incomplete or inconsistent clinical records were excluded from the analysis. Thus, our whole study cohort consisted of 450 patients having a median age of 78 (68–84) years (Figure 1). Males made up 137 (30.4%) of the cohort.

Table 1 shows the demographic data of our patients, divided into the three defined age subgroups. As expected, older patients had more comorbidities, according to their higher CCI, and females were better represented, probably due to the higher life expectancy. Interestingly, the older patients had clinical presentations slightly different from younger patients, complaining more often of dyspnea and less frequently of abdominal pain (Table 1).

### 3.1. In-Hospital Death Occurrence

Overall, 63 (14%) patients died. By univariate analysis, age was significantly correlated with a higher risk of in-hospital death (*p* < 0.001) (Table 2).

Patients’ age, dyspnea presentation at the ED, serum creatinine, WBC, serum albumin, and comorbidities were associated with death occurrence in the univariate analysis. The best cutoff values discriminating the risk of death since ED admission were age > 81 years, serum creatinine > 2.5 mg/dL, WBC > 13.31 × 10^9^/L, and serum albumin ≤ 30 µmol/L. Similarly, the number of comorbidities, expressed by a CCI > 3, was also associated with an increased risk of death. Figure 2 shows the ROC analysis results for age, creatinine, albumin, WBC, and the CCI for all-cause in-hospital death. Interestingly, the qSOFA score at ED admission, which is widely used for risk stratification of infective patients in the emergency setting, was not significantly different between the two groups (Table 2).

No differences were found between the clinical symptoms at ED admission, except for dyspnea, reported by 16 (25.4%) patients who died, compared to 36 (9.3%) patients who survived (*p* < 0.001).

The multivariate analysis revealed that age > 81 years, low serum albumin, high creatinine, and high WBC at ED admission were associated with an increased risk of death (Figure 3). Both CCI and age group were not independently associated with death in our cohort.

### 3.2. Cumulative Major Complication Occurrence

Overall, 105 patients experienced major cumulative complications (Table 3). In the univariate analysis, age was significantly associated with a higher risk of major cumulative complications, (*p* < 0.001). Both the ATLAS and CCI were also significantly associated with major complications (Table 3). 

In terms of in-hospital death, the patients experiencing major complications were older and had higher creatinine levels and higher WBC counts at ED admission. Interestingly, the albumin value and the number of comorbidities were not significantly associated with major complications once adjusted for significant clinical covariates (Table 3).

### 3.3. Overall LOS and ED Readmission within 60 Days

Overall LOS was significantly correlated with age (*p* < 0.05, Table 1) and occurrence of major complications (*p* < 0.001, Table 3), while ED readmission within 60 days did not show a significant difference among the three age subgroups (Table 1), nor with the occurrence of major complications (Table 3).

## 4. Discussion

The major findings of the present work are that in patients with CDI, the risk of in-hospital death and major complications could be effectively predicted upon ED admission. Interestingly, and contrary to expectation, the presence of multiple comorbidities alone is not associated with a different risk of poor outcomes; for example, patients > 81 years old have an increased risk of death and major complications, independent of other clinical covariates.

CDI represents a burdensome clinical issue; it is associated with significant morbidity and mortality worldwide, with a relevant impact on healthcare costs, mainly linked to prolonged hospitalization and rehospitalization [1,2]. Therefore, early identification of patients at risk appears fundamental to optimizing treatment and selecting those patients who might benefit from more aggressive strategies.

Until now, a series of typical risk factors for a more severe and complicated course of CDI have been identified, such as age ≥ 80 years, white cell count of <4 or ≥20 × 10^9^/L, rise in serum creatinine (1.5-fold higher than the premorbid level, or absolute value of 1.5 mg/dL), increase in blood urea nitrogen (>7 mmol/L), and C-reactive protein levels ≥ 150 mg/L [3,4,5,6,7,8,9,10,16]. In particular, advanced age, widely recognized as a key unfavorable prognostic element in most diseases, may contribute by different mechanisms, including the remodeling of the immune system, the higher prevalence of malnutrition and sarcopenia, and the increased occurrence of multimorbidity [13]. In the case of CDI, it has been reported that the mortality rate significantly increases with advancing age [17,18,19], ranging from 3.4% in patients aged <40 years to 41% in those aged >90 years [20].

The present study suggests that, in the emergency setting, older age, together with values of white blood cell count and serum creatinine at ED admission, are independent risk factors for mortality and major cumulative complications.

Moreover, the covariate-adjusted analysis demonstrates that the simple increase in comorbidities is not an independent predictor of poor outcomes. Comorbidities are often considered risk factors in patients with CDI. However, the heterogeneity of the evaluated cohorts, and the small sample sizes considered [21,22], might have limited a correct understanding of its real significance. Lee et al. showed that a different antibiotic strategy based on Charlson’s score, that is, oral metronidazole with a score < 5 or oral vancomycin with a score ≥ 5, might positively influence the survival of CDI patients [7]. However, the CDI group was composed of only 42 subjects, with a quite low median age of 66 years old, and no group included patients aged over 80. A larger cohort study on 2761 CDI patients with a median age of 82.1 years [23] showed that 30-day mortality was predicted by the presence of comorbidities (cancer, cognitive impairment, cardiovascular, respiratory, and kidney disease). As a result, the authors extrapolated a predictive score based on age, renal disease, and cancer (ARC score) that was able to differentiate between groups of patients with varying risk at 30-day absolute mortality. As the same authors underlined, however, this score might have deserved further validation in younger or specialized cohorts, compared with “gold standard” predictors, which have not been available until now [23].

In the present cohort, the clinical presentation of the patients was poorly associated with the prognosis, with the single exception of the presence of dyspnea at ED admission. However, less than 12% of the patients presented this symptom in the study cohort. These data confirm the results of a recent narrative review suggesting the limited association between typical “abdominal” symptoms and CD infection and outcomes, particularly in the elderly patients accessing the ED [24].

Similarly, the qSOFA score, which is commonly used in the emergency setting for the prognostic stratification of patients with infective diagnosis, was not associated with poor outcomes in our cohort. These results can be explained by the relatively mild clinical presentation of most CD infections in the first phase, with a progression to overt sepsis only in the late phases of the disease. This clinical course could be different in patients with immunodepression or malignancy, as demonstrated by a recent report on a small cohort of CDI patients with concomitant solid tumors, in which a qSOFA score ≥ 2 was found to be associated with a poor prognosis [25].

Contrary to clinical signs, some laboratory values at ED admission could have relevant implications for the risk stratification of patients with CDI. In most of the previous studies, and in the present cohort, a high WBC count represented the most reliable marker of severe disease and poor prognosis, although this value has very little specificity [4,7,8,9,10,21,22,23,24]. Similarly, the impairment of renal function, as assessed by elevated serum creatinine, was demonstrated to be associated with a worse prognosis in CDI, such as other infective diseases [4,7,8,9,10,21,22,23,24]. Particular attention should be given to bowel protein loss and poor nutritional status in these patients, since reduced albumin was found to be associated with poor prognosis in several studies, and in the present cohort [7,8,9,10,21,22,23,24]. A systematic performance tool or a validated prognostic score for the early phases of hospitalization or ED admission could help physicians in modulating disease management; for example, knowing when adopt a more aggressive therapeutic approach and/or increase monitoring in selected patients. The ATLAS score (consisting of the variables age, temperature, leukocyte count, albumin, systemic antibiotics, and serum creatinine) was proposed by Miller et al. for the risk stratification of CDI patients [7], and it has been validated in both community and hospital settings [4,8,9,10,16,21,22,23]. However, only a few studies have evaluated CDI in such a complex setting as the ED, although this context deserves particular attention. A recent 8-year project conducted by the Nationwide Emergency Department reported over 900,000 visits to the ED for CDI, and it demonstrated that about 10% of patients visiting the ED for diarrhea then received a diagnosis of CDI [26].

Until now, the few data available on prognostics scores for CDI in the emergency setting could only be derived from studies on ICU patients [20,21]. Sabau et al. showed that the SOFA score at the time of CDI onset correlated with a more complicated course of CDI, together with low levels of serum albumin and the simplified acute physiology (SAPS) score [27]. Conversely, Aguilar et al. found that the ATLAS score failed to predict the cure rate in the ICU setting [28], most likely because this score is validated for a moderately ill population, and is not specifically designed for acutely ill patients.

People visiting the ED, such as our study cohort, represent a heterogeneous group of subjects complaining of mild to severe symptoms. Although selected for hospitalization, our population was quite different from typical ICU patients. It is not surprising, therefore, that a specific disease score for sepsis (qSOFA) was poorly effective for the prognostic stratification of the patients. Interestingly, dyspnea was an independent predictive symptom of poor outcomes in our cohort, possibly for its association with the risk of an underlying septic status.

The present study confirms that old age, and particularly in those past their 8th decade, is a strong risk factor for poor prognosis, with the CDI patients over 81 years in our cohort having a threefold risk of death compared to younger patients, after adjusting for clinical covariates. Most of the current research focuses on the presence of multiple comorbidities to explain the disproportionate mortality of this subgroup of patients [20,21,22,23]. As expected, in the present cohort, older subjects had a higher number of comorbidities, and were more likely to suffer from cardiovascular and pulmonary diseases, and dementia. However, when adjusted for clinical covariates, our analysis revealed that the comorbidities themselves were not associated with a worse prognosis. It is well known in current geriatric research that comorbidities, such as chronological age, do not always truly reflect the overall health status of older patients [28]. For this reason, the “frailty” concept was introduced to better characterize the declined physiologic function, diminished strength, and reduced resilience to stressors that lead to an increased risk of adverse outcomes [28]. The concept of frailty is often confused in clinical practice with multimorbidity. Despite these conditions sharing several aspects, and the fact that chronic diseases are often a key component of the frailty status, the two concepts are clinically distinct [29,30,31,32]. The present research did not include a frailty evaluation, upon admission into the ED, of the enrolled CDI patients; thus, we cannot conclude whether old age is associated with a worse prognosis due to an increase in frailty or due to other clinical reasons, such as the remodeling of the immune system, and the higher prevalence of malnutrition and sarcopenia [13,17,18,19,20].

## 5. Study Limitations

Some limitations are worth considering. Firstly, the single-center retrospective design and the selection of hospitalized patients could lead to the exclusion of a subgroup of CDI with milder infection.

Moreover, we could not evaluate CDI recurrence after the first 60 days, or the multiple CDI recurrence. Prediction of those patients at increased risk of recurrent CDI might guide physicians in the early selection of the candidates for new drugs or strategies, including fecal microbiota transplantation. Finally, we did not perform a separate analysis based on specific antibiotic CDI therapy, because we only selected patients who were hospitalized, and consequently, all were treated according to international guidelines. Further studies should be performed to recognize, upon first presentation in the ED, patients at risk of poor prognosis, including the risk of CDI recurrence.

## 6. Conclusions

In conclusion, early identification of outcome predictors of CDI patients, starting from ED admission, could help to identify the most appropriate management strategy for each patient. Patients in their 8th decade and over are at increased risk of worse prognosis, including major complications and death, regardless of pre-existing comorbidities. Low albumin, increased WBC count and creatinine level, and dyspnea at presentation could be considered “red flags” for patients at increased risk following admission into the ED.

## Figures and Tables

**Figure 1 jpm-12-01573-f001:**
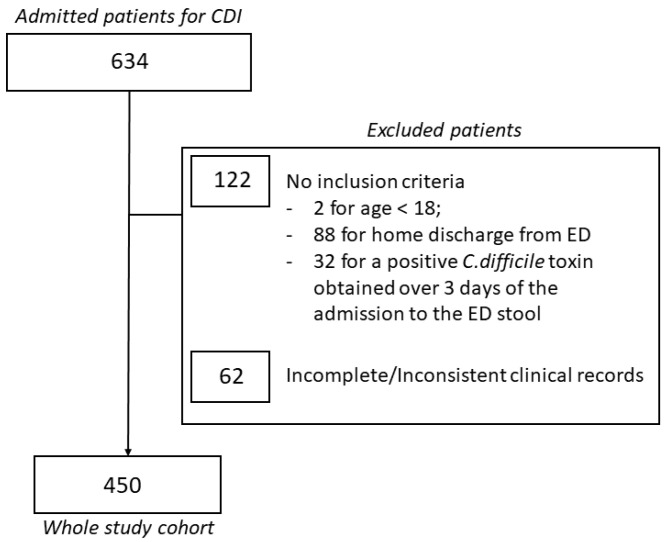
Enrollment process and final study population.

**Figure 2 jpm-12-01573-f002:**
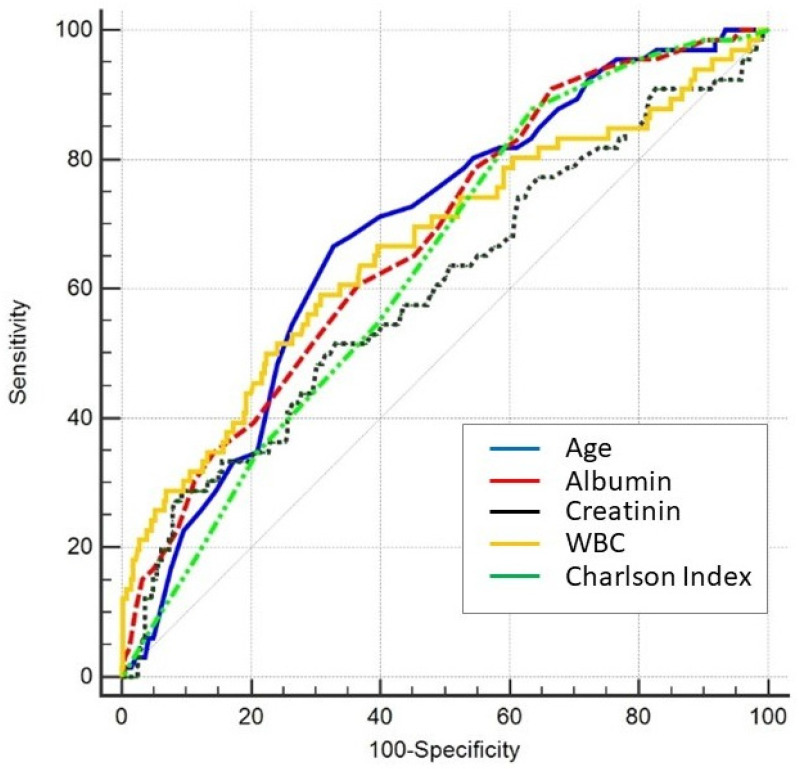
Receiver operating characteristic (ROC) analysis for age, creatinine, albumin, white blood cell count (WBC), and Charlson comorbidity index (CCI) at admission for all-cause in-hospital death. ROC areas under the curve: age, 0.684 (0.640–0.726); creatinine, 0.675 (0.631–0.717); WBC, 0.665 (0.620–0.707); albumin, 0.675 (0.631–0.717); and CCI 0.637 (0.592–0.681).

**Figure 3 jpm-12-01573-f003:**
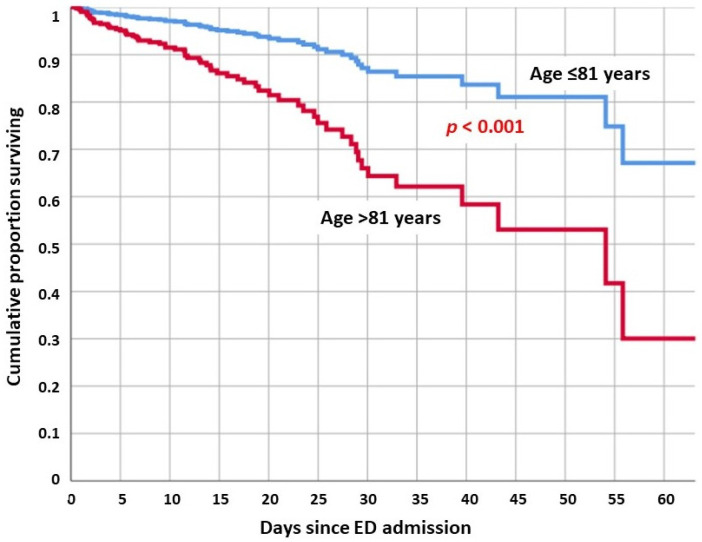
Adjusted cumulative survival of CDI patients according to age divided by the cutoff value of 81 years indicated by the ROC analysis Youden index (J). Adjusted mortality was about threefold higher in older patients regardless of comorbidities and other clinical covariates.

**Table 1 jpm-12-01573-t001:** Population demographics.

Variable	Age 19–64 N = 80	Age 65–79 N = 171	Age > 80 N = 199	*p*
Median Age	53 (43–61)	74 (69.5–77)	85 (82–89)	
Sex (Male)	27 (33.8%)	62 (36.3%)	48 (24.1%)	0.032
*ED Presentation and not Ed presentation*				
Antibiotic therapy	74 (92.5%)	155(90.6%)	180 (90.4%)	0.418
Previous clostridium infection	16 (20.0%)	51 (29.8%)	51 (25.6%)	0.249
Fever	42 (52.5%)	58 (33.9%)	71 (35.7%)	0.012
Abdominal pain	29 (36.3%)	24 (14.0%)	35 (17.6%)	<0.001
Vomit	9 (11.3%)	19 (11.1%)	21 (10.6%)	0.979
Dyspnea	2 (2.5%)	25 (14.6%)	25 (12.6%)	0.017
Syncope	3 (3.8%)	7 (4.1%)	9 (4.5%)	0.953
Malaise/fatigue	8 (10.0%)	20 (11.7%)	24 (12.1%)	0.886
Gastrointestinal bleeding	4 (5.0%)	4 (2.3%)	4 (2.0%)	0.354
qSOFA score	0 (0–1)	0 (0–1)	0 (0–1)	0.422
*Laboratory Values*				
Serum Creatinine mg/dL	0.76 (0.55–1.16)	0.91 (0.62–1.47)	1.22 (0.78–1.83)	<0.001
White blood cell count	10.02 (7.33–14.32)	10.6 (7.74–15.90)	11.5 (7.95–17.01)	0.163
Serum Albumin	30.5 (24.7–34)	27 (23–30)	26 (22–31)	0.003
*Comorbidities*				
Charlson Comorbidity Index	1 (0–2)	4 (3–5)	5 (4–6)	<0.001
Ischemic heart disease	7 (8.8%)	19 (11.1%)	28 (14.1%)	0.420
Congestive heart failure	4 (5.0%)	19 (11.1%)	41 (20.6%)	0.001
Peripheral vascular disease	8 (10.0%)	36 (21.1%)	50 (25.1%)	0.019
Dementia	0	11 (6.4%)	26 (13.1%)	0.001
COPD	0	25 (14.6%)	33 (16.6%)	0.001
Diabetes	7 (8.8%)	21 (12.3%)	33 (16.6%)	0.185
Chronic kidney disease	2 (2.5%)	12 (7.0%)	23 (11.6%)	0.034
Rheumatologic disease	0	3 (1.8%)	2 (1.0%)	0.458
HIV infection	1 (1.3%)	1 (0.6%)	1 (0.5%)	0.775
Leukemia/Lymphoma	3 (3.8%)	6 (358%)	6 (3.0%)	0.941
Solid Malignancy	7 (8.8%)	19 (11.1%)	9(4.5%)	0.058
*Outcomes*				
Death	3 (3.8%)	15 (8.8%)	45 (22.6%)	<0.001
Mechanical Ventilation	2 (2.5%)	7 (4.1%)	4 (2.0%)	0.478
Sepsis	9 (11.3%)	24 (14.0%)	37 (18.6%)	0.243
Colon surgery	0	1 (0.6%)	1 (0.5%)	0.799
Major Cumulative Complications ^#^	11 (13.8%)	32 (18.7%)	62 (31.2%)	0.002
Length of Hospital Stay (days)	6.1 (3.1–14.5)	12.1 (6.8–19.9)	9.9 (5.5–16.9)	0.001
ED readmission in 60 days	7 (8.8%)	17 (9.9%)	15 (7.5%)	0.715

^#^ Major cumulative complications include death, admission to ICU/ventilation, sepsis, and colon surgery.

**Table 2 jpm-12-01573-t002:** The univariate and multivariate comparison regarding all-cause in-hospital death.

Variable	Survived	Deceased	Univ. *p* Value	Hazard Ratio (95% CI)	Multiv. *p* Value
Median cumulative age	77 (67–83)	83 (78–87)			
Age > 81 ^§^ years	133 (32.8%) ^§^	44 (66.7%)	**<0.001**	3.02 (1.64–5.53)	**<0.001**
Sex (Male)	122 (31.5%)	15 (23.8%)	0.240		
*ED Presentation and not Ed presentation*					
Antibiotic therapy	357 (90.9%)	52 (92.4%)	0.684		
Previous clostridium infection	99 (25.6%)	19 (30.2%)	0.443		
Fever	145 (37.5%)	26 (41.3%)	0.578		
Abdominal pain	77 (19.9%)	11 (17.5%)	0.734		
Vomit	39 (10.1%)	10 (15.9%)	0.189		
Dyspnea	36 (9.3%)	16 (25.4%)	**<0.001**	1.79 (1.01–3.17)	**0.047**
Syncope	18 (4.7%)	1 (1.6%)	0.495		
Malaise/fatigue	44 (11.4%)	8 (12.7%)	0.831		
Gastrointestinal bleeding	10 (2.6%)	2 (3.2%)	0.679		
qSOFA score	0 [0,1]	0 (0–1)	0.183		
*Laboratory Values*					
Serum Creatinine mg/dL	0.94 (0.67–1.52)	1.41 (0.81–2.80)	0.002		
Creatinine > 2.5 ^§^ mg/dL	38 (9.4%)	19 (28.8%)	**<0.001**	1.88 (1.07–3.29)	**0.028**
WBC (Cells × 10^9^/L)	10.7 (7.58–15.29)	15.7 (10.44–24.36)	<0.001		
WBC > 13.31 ^§^ (Cells × 10^9^/L)	125 (30.8%)	39 (59.1%)	**<0.001**	1.74 (1.03–2.94)	**0.037**
Serum Albumin (µmol/L)	27 (24–32)	24 (20–28)	<0.001		
Albumin ≤ 30 ^§^ (µmol/L)	270 (66.5%)	60 (90.9%)	**<0.001**	2.95 (1.25–6.93)	**0.013**
*Comorbidities*					
Charlson Comorbidity Index	4 (3–5)	5 (4–6)	<0.001		
Charlson Index > 3 ^§^	258 (63.5%)	58 (87.9%)	**<0.001**	1.95 (0.73–5.18)	0.181
Ischemic heart disease	48 (12.4%)	6 (9.5%)	0.676		
Congestive heart failure	50 (12.9%)	14 (22.2%)	0.077		
Peripheral vascular disease	80 (20.7%)	14 (22.2%)	0.741		
Dementia	29 (7.5%)	8 (12.7%)	0.211		
COPD	44 (11.4%)	14 (22.2%)	0.025		
Diabetes	48 (12.4%)	13 (20.6%)	0.109		
Chronic kidney disease	31(8.0%)	6 (9.5%)	0.626		
Rheumatologic disease	5 (1.3%)	0	1.000		
HIV infection	3 (0.8%)	0	1.000		
Leukemia/Lymphoma	13 (3.4%)	2 (3.2%)	1.000		
Solid Malignancy	29 (7.5%)	6 (9.5%)	0.610		
*Outcomes*					
Length of Hospital Stay (days)	9.5 (5.0–17.1)	11.5 (3.8–21.0)	0.676		

Abbreviations: WBC—white blood cell; qSOFA—quick sequential organ failure assessment; COPD—chronic obstructive pulmonary disease. ^§^ Cutoff values were chosen according to the Youden index (J) in ROC analysis for death.

**Table 3 jpm-12-01573-t003:** Univariate and multivariate analysis for major cumulative complications ^#^.

Variable	None or Minor Complications	Cumulative Major Complications	Univariate *p* Value	Hazard Ratio (95% CI)	Multiv. *p* Value
Median cumulative age	77 (67–83)	82 (72–86)	<0.001		
Age > 81 ^§^ years	116 (32.0%)	61 (55.5%)	**<0.001**	2.90 (1.77–4.75)	**<0.001**
Sex (Male)	111 (32.2%)	26 (24.8%)	0.183		
*ED Presentation and not Ed presentation*					
Antibiotic therapy	317 (90.3%)	92 (93.6%)	0.286		
Previous clostridium infection	87 (25.2%)	31 (29.5%)	0.378		
Fever	126 (36.5%)	45 (42.9%)	0.252		
Abdominal pain	74 (21.4%)	14 (13.3%)	0.069		
Vomit	34 (9.9%)	15 (14.3%)	0.212		
Dyspnea	29(8.4%)	23 (21.9%)	**0.001**	1.45 (0.89–2.34)	0.130
Syncope	15 (4.3%)	4 (3.8%)	1.000		
Malaise/fatigue	36 (10.4%)	16 (15.2%)	0.221		
Gastrointestinal bleeding	9 (2.6%)	3 (2.9%)	1.000		
qSOFA score	0 (0–1)	0 (0–1)	0.184		
*Laboratory Values*					
Serum Creatinine mg/dL	0.92 (0.68–1.49)	1.21 (0.74–2.25)	0.012		
Creatinine > 2.5 ^§^ mg/dL	33 (9.1%)	24 (21.8%)	**<0.001**	1.64 (1.01–2.67)	**0.047**
WBC (Cells × 10^9^/L)	10.65 (7.61–14.94)	14.02 (9.01–21.2)	<0.001		
WBC > 13.31 ^§^ (Cells × 10^9^/L)	107 (29.6%)	57 (51.8%)	**<0.001**	1.61 (1.07–2.42)	**0.022**
Serum Albumin (µmol/L)	28 (24–32)	25 (21–29)	0.001		
Albumin ≤ 30 ^§^ (µmol/L)	242 (66.9%)	88 (80.0%)	**0.008**	1.45 (0.88–2.37)	0.141
*Comorbidities*					
Charlson Comorbidity Index	4 (3–5)	5 (4–6)	0.002		
Charlson Index > 3 ^§^	233 (64.4%)	83 (75.5%)	**0.030**	1.13 (0.60–2.14)	0.701
Ischemic heart disease	43 (12.5%)	11 (10.5%)	0.732		
Congestive heart failure	40 (11.8%)	24 (22.9%)	0.006		
Peripheral vascular disease	70 (20.3%)	24 (22.9%)	0.585		
Dementia	25 (7.2%)	12 (11.4%)	0.221		
COPD	37 (10.7%)	21 (20.0%)	0.019		
Diabetes	45 (13.0%)	16 (15.2%)	0.625		
Chronic kidney disease	28 (8.1%)	9 (8.6%)	0.841		
Rheumatologic disease	5 (1.4%)	0	0.595		
HIV infection	2 (0.6%)	1 (1.0%)	0.550		
Leukemia/Lymphoma	9 (2.6%)	5.7 (7.3%)	0.128		
Solid Malignancy	24 (7.0%)	11 (10.5%)	0.296		
*Outcomes*					
Length of Hospital Stay (days)	8.8 (4.7–14.1)	16.0 (6.7–30.5)	<0.001		
Readmission in ED in 60 days	34 (9.9%)	5 (4.8%)	0.116		

Abbreviations: WBC—white blood cell; qSOFA—quick sequential organ failure assessment; COPD—chronic obstructive pulmonary disease. ^§^ Cutoff values are chosen according to Youden index (J) in ROC analysis with respect to death. # Major cumulative complications include death, admission to ICU/ventilation, sepsis, colon surgery.

## Data Availability

Non applicable.

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
