# Peer review of "Early Prognostic Stratification of Clostridioides difficile Infection in the Emergency Department: The Role of Age and Comorbidities"

_jpm, 2022, doi:10.3390/jpm12101573_

Round 1
Reviewer 1 Report
1. Clostridioides difficile and C difficile should be italicized throughout the manuscript.
2. Line 21. It is not clear what the authors meant by "these ones except for low albumin". If low albumin was not associated with death AND major complications, then low albumin should be removed from this statement.
3. Line 57. I don't think "poor is known" is a meaningful statement. Perhaps the authors meant: little is known?
4. Line 76. Perhaps the authors meant: obtained within 3 days of the admission to the ED.
5. Line 103. What was the definition of sepsis. Did the authors mean sepsis based on SIRS criteria?
6. Lines 214, 242, and 277. What does "Since now" mean?
7. Lines 242. What does sinc
Major:
1. The authors' primary outcome was the combination of death AND major complications, yet when analyzing data there was a lot emphasis on death as a separate outcome. There was table 3 and figure 1 and 2 discussing death. It was confusing to read the manuscript when there was a separate paragraph on death (3.1) and another paragraph (3.2) for the cumulative major complication, which included death AND major complications. I think it is best to better define what the primary end points are. If the authors would like to address all-cause in-hospital death (mortality) separately, then they should better define that in paragraph 2.3. For example primary outcome measure: 1. All-cause in-hospital death and 2. The cumulative major complications which included: all-cause in-hospital death, admission to ICU/ventilation, sepsis, colon surgery.
In summary, I believe it is best to have paragraph 2.3 consistent with they way paragraphs 3.1, 3.2 and 3.3 are addressed in the manuscript.
2. Line 225: Authors state that the value of albumin at ED admission is and independent risk factor for death and cumulative major complications. However, in table 3. the albumin was not a predictive of the cumulative major complications in the multivariate analysis. I think this is related to the ambiguous definition of cumulative major complications.
Author Response
1.Clostridioides difficile and C difficile should be italicized throughout the manuscript.
They were italicized
- Line 21. It is not clear what the authors meant by "these ones except for low albumin". If low albumin was not associated with death AND major complications, then low albumin should be removed from this statement.
We agree with the Referee. As the sentence resulted confused, we removed the statement regarding low albumin levels.
- Line 57. I don't think "poor is known" is a meaningful statement. Perhaps the authors meant: little is known?
Thanks for the suggestion. We changed the statement
- Line 76. Perhaps the authors meant: obtained within 3 days ofthe admission to the ED.
Thanks for the suggestion. We changed the statement
- Line 103. What was the definition of sepsis. Did the authors mean sepsis based on SIRS criteria?
In this work we considered sepsis according to the more recent criteria elaborated by the “Third International Consensus Definitions for Sepsis and Septic Shock (Sepsis-3)”. This reference was added in the text (reference 16)
- Lines 214, 242, and 277. What does "Since now" mean?
We apologize. The term “since now” was changed with “till now” in every of the mentioned parts of the manuscript
- Lines 242. What does sinc
See response to point 6
Major:
- The authors' primary outcome was the combination of death AND major complications, yet when analyzing data there was a lot emphasis on death as a separate outcome. There was table 3 and figure 1 and 2 discussing death. It was confusing to read the manuscript when there was a separate paragraph on death (3.1) and another paragraph (3.2) for the cumulative major complication, which included death AND major complications. I think it is best to better define what the primary end points are. If the authors would like to address all-cause in-hospital death (mortality) separately, then they should better define that in paragraph 2.3. For example primary outcome measure: 1. All-cause in-hospital death and 2. The cumulative major complications which included: all-cause in-hospital death, admission to ICU/ventilation, sepsis, colon surgery.
In summary, I believe it is best to have paragraph 2.3 consistent with they way paragraphs 3.1, 3.2 and 3.3 are addressed in the manuscript.
We totally agree and thank the Referee’s for his comment. We apologized for the confusion derived from a not completely correspondence between outcome descriptions and legend of table relative to major complications, which effectively also included in-hospital death. We modified the paragraph relative to outcomes in Patients and Methods sections, better describing the primary outcomes.
- Line 225: Authors state that the value of albumin at ED admission is and independent risk factor for death and cumulative major complications. However, in table 3. the albumin was not a predictive of the cumulative major complications in the multivariate analysis. I think this is related to the ambiguous definition of cumulative major complications.
The Referee is right. The low albumin levels were correlated with mortality but not with cumulative major complications. We changed the sentence, deleting the data relative to albumin.
Reviewer 2 Report
This is an interesting study, but I advise the authors to clarify its novelty in the Introduction . A flowchart of the study should also be provided, as the numbers of screened, excluded, and included persons do not match.
Author Response
This is an interesting study, but I advise the authors to clarify its novelty in the Introduction . A flowchart of the study should also be provided, as the numbers of screened, excluded, and included persons do not match.
We thank the Reviewer for his appreciation and his comments. We better underlined the novelty of our study in the Introduction Section and corrected numbers of screened, excluded and included person in the Results Section. We also prepare a figure, that should be numbered as Figure 1 since it appears firstly in the Result section, representing the flow-chart of enrollment process. Therefore, the other figures were renumbered.
